# Serratiopeptidase Attenuates *Lipopolysaccharide*-Induced Vascular Inflammation by Inhibiting the Expression of Monocyte Chemoattractant Protein-1

Vikas Yadav [1] , Satyam Sharma [1] , Ashutosh Kumar [2,3], Sanjiv Singh [1,*] and V. Ravichandiran [1]

1 Department of Pharmacology and Toxicology, National Institute of Pharmaceutical Education and Research, Hajipur 844102, Bihar, India
2 Department of Pharmacology and Toxicology, National Institute of Pharmaceutical Education and Research, Kolkata 700054, West Bengal, India
3 Department of Pharmacology and Toxicology, National Institute of Pharmaceutical Education and Research, Sahibzada Ajit Singh Nagar 160062, Punjab, India
* Correspondence: hodpharmacology@niperhajipur.ac.in or sanjivsingh@niperhajipur.ac.in; Tel.: +91-6224277226

**Abstract:** Lipopolysaccharide (LPS) has potent pro-inflammatory properties and acts on many cell types including vascular endothelial cells. The secretion of the cytokines MCP-1 (CCL2), interleukins, and the elevation of oxidative stress by LPS-activated vascular endothelial cells contribute substantially to the pathogenesis of vascular inflammation. However, the mechanism involving LPS-induced MCP-1, interleukins, and oxidative stress together is not well demonstrated. Serratiopeptidase (SRP) has been widely used for its anti-inflammatory effects. In this research study, our intention is to establish a potential drug candidate for vascular inflammation in cardiovascular disorder conditions. We used BALB/c mice because this is the most successful model of vascular inflammation, suggested and validated by previous research findings. Our present investigation examined the involvement of SRP in vascular inflammation caused by lipopolysaccharides (LPSs) in a BALB/c mice model. We analyzed the inflammation and changes in the aorta by H&E staining. SOD, MDA, and GPx levels were determined as per the instructions of the kit protocols. ELISA was used to measure the levels of interleukins, whereas immunohistochemistry was carried out for the evaluation of MCP-1 expression. SRP treatment significantly suppressed vascular inflammation in BALB/c mice. Mechanistic studies demonstrated that SRP significantly inhibited the LPS-induced production of proinflammatory cytokines such as IL-2, IL-1, IL-6, and TNF-α in aortic tissue. Furthermore, it also inhibited LPS-induced oxidative stress in the aortas of mice, whereas the expression and activity of monocyte chemoattractant protein-1 (MCP-1) decreased after SRP treatment. In conclusion, SRP has the ability to reduce LPS-induced vascular inflammation and damage by modulating MCP-1.

**Keywords:** serratiopeptidase; vascular inflammation; lipopolysaccharides; monocyte chemoattractant protein-1; oxidative stress; vascular smooth muscle cells

## 1. Introduction

Heart diseases (CVDs) arise primarily from vascular endothelial abnormalities and are the leading cause of mortality globally [1]. Only non-steroidal and steroidal anti-inflammatory drugs are now used as treatments for inflammatory illnesses. The chronic usage of these medications has been linked to serious negative side effects such as kidney, cardiac, and digestive disturbances. New anti-inflammatory drugs with specific actions and lower toxicity are urgently needed. New anti-inflammatories are likely to come from natural sources, which is exciting and encouraging. Despite the numerous anti-inflammatory drug candidates for the development of potential therapeutic approaches are missing. The careful selection of the most suitable therapy is a crucial step in the selection

of the best anti-inflammatory drug candidate. An organized assessment of in vivo animal models can make it easier to find and develop possible anti-inflammatory leads from nature [2]. In a resting condition, the propensity for tissue death and the protective role of the inflammation response are typically balanced. Contrarily, persistent inflammation is typically characterized by significant tissue damage and repair following an inflammatory process [3]. Due to the persistence of low-grade, persistent inflammation, many serious diseases appear. Due to the absence of safe and efficient medications, treating chronic inflammatory disorders such as rheumatoid arthritis, inflammatory bowel diseases, and endothelium dysfunction remains difficult [4].

In addition, Tacke et al. showed that MCP-1 enhanced hepatic monocyte accumulation. Novel anti-MCP-1-directed drugs can selectively prevent the entry of pro-inflammatory monocytes into damaged mouse livers. MCP-1 is therefore essential for the hepatic inflammatory response. In the present work, we showed that propofol dramatically increased the expression of apoM and foxa2 in HepG2 cells, which inhibited LPS-induced MCP-1 production. The subfamily cytokine monocyte chemoattractant protein-1 (MCP-1), also known as CCL2, is a model member and a crucial player in the start of inflammation. Numerous cell types, including fibroblasts, macrophages, endothelium, epithelial, vascular smooth muscle, and other cell types that control the flow of cells to inflammatory areas, generate MCP-1. It interacts with the membrane-bound CC chemokine receptor type 2 (CCR2) of monocytes to cause chemotaxis and the transepithelial migration of monocytes to inflammatory areas. MCP-1 expression is increased in response to tissue damage and pro-inflammatory stimuli. Numerous studies have shown that MCP-1 is expressed in a variety of tissue types when atherosclerosis, arthritis, and cancer develop and evolve as inflammatory diseases. In these situations, it has been hypothesized that the invasion of macrophages by tissues will make the disorders worse. Therefore, MCP-1 expression has to be carefully controlled since it is anticipated to be essential for battling inflammatory disorders [5].

The powerful monocyte attractant monocyte chemoattractant protein (MCP)-1, also known as chemokine (C-C motif) ligand 2 (CCL2), is a member of the CC chemokine subfamily. Chemokines are well established to play a significant functional role in promoting immune system cell migration and activation. Importantly, chemokines encourage the directed migration of inflammatory cells, which plays a role in the development of atherosclerosis. MCP-1 makes it easier for macrophages to be drawn into the vascular intima, where they help control other signaling pathways important for the development of arteriosclerosis and metabolic abnormalities. Therefore, MCP-1 expression has to be carefully controlled since it is anticipated to be essential for battling inflammatory disorders [6]. LPS directly stimulates the endothelial cells and causes a variety of EC reactions, such as an increase in the expression of particular binding proteins and cytokine production such as IL-1, IL-8, and MCP-1 (monocyte chemotactic protein-1), which in turn causes the preferential recruitment of immune cells to inflammatory foci [7].

MCP-1 also promotes the production of metalloproteinase, tissue factor, chemokines, pro-inflammatory cytokines, and adhesion molecules. These results suggest that MCP-1 has a role in the onset and progression of inflammation when taken as a whole. It is widely understood that inflammation caused by a range of factors such as pro-inflammatory mediators and specific bacterium causes rapid and significant leukocyte recruitment in the pathological mechanism of the damaged endothelium and might worsen endothelial cellular damage as well as initiate the cardiovascular process [8–11]. The vasculature is a permeable membrane in between the vessel walls and the circulatory system. It is accountable for keeping circulatory flowability and governing the circulation of blood in noninflamed tissues, as well as controlling normal vascular penetration and quiescing circulating leukocytes [12–14]. Injury results in the development and migration of vascular smooth muscle cells (VSMCs), the proliferation of endothelial cells, and the activation of macrophages in the intima layer of blood vessels. All of these things cause neointimal hyperplasia and an increase in medium wall thickness. Vascular remodeling occurs as

a result of oxidative stress, inflammation, and the pathophysiology of various disorders. Additionally, it was confirmed that vascular lesions had higher levels of NADPH oxidase, a significant generator of free radicals in the artery wall. Ultimately, maintaining vascular homeostasis depends on the cellular defenses against oxidative stress and inflammation. Complete carotid ligation causes neointimal hyperplasia and flow-induced vascular remodeling as a result of cutting off blood flow near the bifurcation. It is crucial to take into account the natural system for inflammation management while thinking about treatment options [15].

Intriguingly, scientific results demonstrated that rivaroxaban pre-treatment in ischemic cardiomyopathy in a mouse model with diet-induced myocardial infarction has cardio-protective effects. One such study demonstrated that lowering the cardiovascular mRNA expression of IL-1, TNF-$\alpha$, and IL-6, as well as in the nuclear factor kappa B (NF$\kappa$B)-activation pathway in the rivaroxaban pre-treated group, is necessary to attenuate cardiac remodeling and fibrosis and alleviate aortic root and coronary artery atherosclerosis. In the myocardial reperfusion damage mouse model, rivaroxaban also increased survival rates and cardiac performance and decreased the cardiac mRNA expression of IL-6, collagen 1$\alpha$2, and 3$\alpha$1. The therapeutic efficacy of RVX has also been demonstrated in a rat model of central nervous system ischemia/reperfusion injury, in which it prevented the expression of the VCAM-1 protein, macrophage activation, and the formation of thrombin-mediated thrombus. Additionally, it was demonstrated in a mouse model of stress overload-induced atrial remodeling with transverse aortic constriction, where it prevented the infiltration of macrophages and the expression of MCP-1, IL-6, and IL-1$\beta$ [16].

One more research finding demonstrated that the equilibrium seen between blood coagulation as well as the anti-platelet system is altered when LPS promotes coagulation and subsequent intravascular coagulation. Additionally, LPS causes the release of a number of inflammatory and chemotactic cytokines from monocytes and endothelial cells, which in turn set off a systemic inflammation response. This interaction between thrombosis and inflammatory response causes the creation of microthrombi, which aid in the emergence of organ dysfunction. The latest disseminated intravascular coagulation outbreak, in particular, draws attention to the fatal consequences of bronchospasm, which is marked by widespread alveolar destruction, pulmonary microvascular endothelial cell damage, and even respiratory failure [17].

Endothelium also serves as one of the initial obstacles against foreign invasion. Physical factors including shear stress and pressure stress as well as biochemical stimuli such as neurotransmitters and locally released vasoactive substances are examples of these invasions. Endothelial cells respond to stimuli by releasing substances that govern hemostasis, cell development, vasomotor functioning, and inflammatory processes. The response is divided into vasodilatation and vasoconstrictors. Nitric oxide (NO), prostacyclin, and endothelial-derived hyperpolarizing agents are all vasodilators. Endothelin-1, angiotensin II (Ang II), and thromboxane A2 are vasoconstrictors [18–20]. Serratiopeptidase (SRP) is created from Serratia microorganisms that reside in the intestines of the silkworm (Bombyx mori), which is why it is usually known as the "silkworm enzyme" [21]. It is a common proteolytic enzyme utilized in clinical uses. It has demonstrated strong anti-inflammatory, anti-epileptic, as well as analgesic benefits in a variety of locations [22]. SRP has also been shown to decrease the production of IL-6, TNF-$\alpha$, and chemokines [23] and has also been seen to be effective against an aluminum chloride-induced Alzheimer's disease animal model [24]. SRP lowers cell surface proteins Ami4b, autolysin, internalin, and ActA in vitro, reducing Listeria monocytogenes' capability to form biofilms as well as enter host tissues This results in the inhibition of Listeria monocytogenes' first adherence to the human intestine [25]. According to some research, SRP in combination with the co-delivery of curcumin with nanostructures improves anti-cancer efficacy in HeLa and MCF-7 cells [26]. Furthermore, statistics indicate that SRP, by its proteolytic, anti-oxidant, and anti-amyloidogenic activities, can downregulate the amyloidogenic pathway. Another recent report also confirmed these findings [27,28]. Lipopolysaccharide (LPS) is

a pro-inflammatory endotoxin that affects a variety of cells, especially the endothelium. Monocyte chemoattractant protein-1 (MCP-1) (CCL2) expression from LPS-stimulated endothelial cells certainly contributes to the pathophysiology of sepsis. Furthermore, the underlying mechanisms of LPS-induced MCP-1 synthesis in endothelial cells are unknown; however, one researcher found that LPS stimulates Pyk2, and p38 mitogen-activated protein kinases are then activated. The transcription factor NF-B is then activated by the p38 Mapk, causing MCP-1 and perhaps other cytokines to be activated transcriptionally [29]. The most critical cytokine that governs monocyte/macrophage movement and invasion is MCP-1. In reaction to cytokines, growth hormones, oxLDL, and CD40L, MCP-1 is significantly elevated in atherosclerotic lesions. MCP-1 was shown to be substantially linked with the quantity of atherosclerosis and macrophage infiltration in animal investigations [30]. The information regarding the effect of SRP on vascular inflammation is definite, as is how the underlying mechanism is related to the MCP-1 pathway. Discovering the processes that control inflammation may aid in the discovery of new treatment targets with significant clinical implications. Thereby, the main objective of the study was to explore the following: (1) the molecular and cellular pathways underlying the protection exerted by SRP against vascular inflammation; and (2) the direction towards recognizing the involvement of MCP-1 in LPS-induced vascular inflammation as an attainable pharmacological target of SRP. Exceptionally, our research studies assessed the hypothesis that SRP decreases both acute inflammation secondary to LPS-induced vascular inflammation by suppressing the MCP-1, oxidative, and inflammatory cascades.

## 2. Materials and Method

### 2.1. Experimental Animals

The Institutional Animal Ethics Committee (IAEC) of the National Institute of Pharmaceutical Education and Research (NIPER) in Hajipur, India, approved the research protocol, and all experiments followed the guidelines set forth by the Committee for the Purpose of Control and Supervision of Experiments on Animals (CPCSEA) in New Delhi, India, IAEC Certificate number NIPER-H/IAEC/09/21.

For the experiments, 12-week-old male BALB/c mice of weight around 30.0–35.0 g were involved. Six animals in each group, with a total of 24 animals, were used. Animals were placed in separate enclosures with regulated temperatures ($22 \pm 3$ °C), humidity (50 20%), a 12 h light–dark cycle, and unlimited food and water. All animals were fasted for 2 hours before and after drug administration. Thirty mice were randomly assigned to one of four distinct treatment groups: vehicle (0.9% normal saline), LPS (1 mg/kg), LPS-SRP (20 mg/kg), and LPS +atorvastatin (20 mg/kg), each with six animals. Drugs were delivered intragastrically once daily for 15 days, and on the final day, normal saline or LPS was injected intraperitoneally 30 min after drug delivery. Blood samples were collected 12 h after injection with normal saline or LPS and then centrifuged at 3500 rpm for 10 min at 4 °C. The blood was taken and kept at −80 °C for the ELISA test. All of the mice were euthanized, and then their thoracic aortas were immediately separated, with some tissues kept in a 4% formaldehyde solution and others at −80° C.

### 2.2. Oxidative Stress Parameters in Aorta

Arterial malondialdehyde (MDA) was identified biologically utilizing spectroscopic analysis based on the previously described reaction between thiobarbituric acid and MDA that also measures the thiobarbituric acid reactive substance concentration, also known as MDA composition. This approach assesses aortic MDA, a byproduct of heart lipid peroxides. Superoxide dismutase (SOD) activity was measured analytically, as reported earlier, and was therefore based on SOD's capacity to prevent pyrogallol autoxidation. One unit of SOD is equivalent to the enzyme level that inhibits 50% of pyrogallol autoxidation; SOD activity was measured at 420 nm [31].

The arterial concentration of reduced glutathione (GSH) was measured using spectroscopic analysis. The approach is based on the fact that the sulfhydryl group of GSH

reacts with Ellman's reagent, 5, 5′-dithio-bis-2-nitrobenzoicacid, to create a yellow-colored 5-thio-2-ni-trobenzoic acid. A Beckman DU-64 UV/VIS spectrophotometer was used to detect GSH at 405 nm [32].

### 2.3. Tumor Necrosis Factor-α and Interleukin in Aorta

Overall concentrations of biomarkers of inflammation in the aorta were determined by ELISA kits and the manufacturer's guidelines. TNF-α (Catalog No. E-EL-R0019, Sigma-Aldrich Co., St. Louis, MO, USA). Interleukins (BMS629) (Thermo Fisher Scientific Inc./Lab. Vision, Waltham, MA, USA).

### 2.4. Immunohistochemistry of Aortic Tissue

Tissues were perfusion-fixed with a mixture of 4% formaldehyde in phosphate-buffered saline (PBS) and were embedded. All frozen sections were cut to 6 μm thick and were permeabilized with 0.1% Triton X-100 in Tris-buffered saline (TBS) for 10 min at room temperature. Non-specific sites were blocked using 1% bovine serum albumin (BSA) and 10% normal donkey serum in TBS for 2 h at room temperature. All primary antibodies diluted in TBS with 1% BSA were then applied onto arterial sections and incubated overnight at 4 °C. On the second day, arterial sections were rinsed with TBS plus 0.025% Triton X-100 with gentle agitation twice, for five minutes each time. Then, the arterial tissues were incubated with fluorophore-conjugated secondary antibodies diluted in TBS with 1% BSA for 1 h at room temperature. Arterial sections were then rinsed with TBS three times for five minutes each time. DAPI was used to stain the nuclei. Staining was visualized with a ZEISS Axiocam Ti inverted microscope system and digital images were acquired [33].

### 2.5. Histological Analysis

All rat aorta tissue was preserved in 4% PFA (paraformaldehyde) over 48 h. Following fixation, this tissue was dehydrated by using an alcohol gradient before being incorporated in paraffin wax liquid. For histopathological investigation, tissue slices were cut at 5 μm thickness and followed by staining with hematoxylin and eosin (H&E). Microscopy was used to examine each segment under 10× light microscopic fields (Olympus Japan Co., Tokyo, Japan).

### 2.6. Statistical Analysis

In this present investigation, the values are provided as the mean ± S.E.M. At $p = 0.05$, variations were considered significant. GraphPad prism (GraphPad Software, San Diego, CA, USA) was used to analyze the results, which included a one-way ANOVA test accompanied by Tukey's test.

## 3. Results

### 3.1. Effect of SRP against Oxidative Stress in Aorta

Total antioxidant capacity (T-AOC) activity, SOD activity, glutathione/glutathione disulfide (GSH/GSSG) levels, and MDA levels were used to measure the oxidative status in order to investigate the role of oxidative stress on the progression of vascular dysfunction in LPS-induced mice and the effects of SRP on oxidative stress on diabetic mice. Cardiac MDA levels were considerably higher in the LPS group versus the control group. Pre-treatment with atorvastatin plus SRP resulted in a substantial decrease in aortic MDA levels. SOD activity, ROS intensity, and GSH levels in the LPS group decreased compared to the control group. SRP alone, but not in combination with LPS, significantly raised cardiac SOD and GSH levels as compared to the LPS group (Figure 1).

### 3.2. Effect on Saortic IL-2, IL-4, IL-6, and TNF-α Levels

When compared to the control group, the LPS group had a substantial rise in aortic TNF-α levels. SRP-treated groups had much lower levels of cardiac TNF-α than LPS-treated

groups. Cardiac TNF levels were considerably higher in the group administered with SRP and LPS than in the treatment with SRP monotherapy. When compared to the control group, the overall aortic IL-2, IL-4, and IL-6 contents in the LPS group were significantly increased (Figure 2A–D).

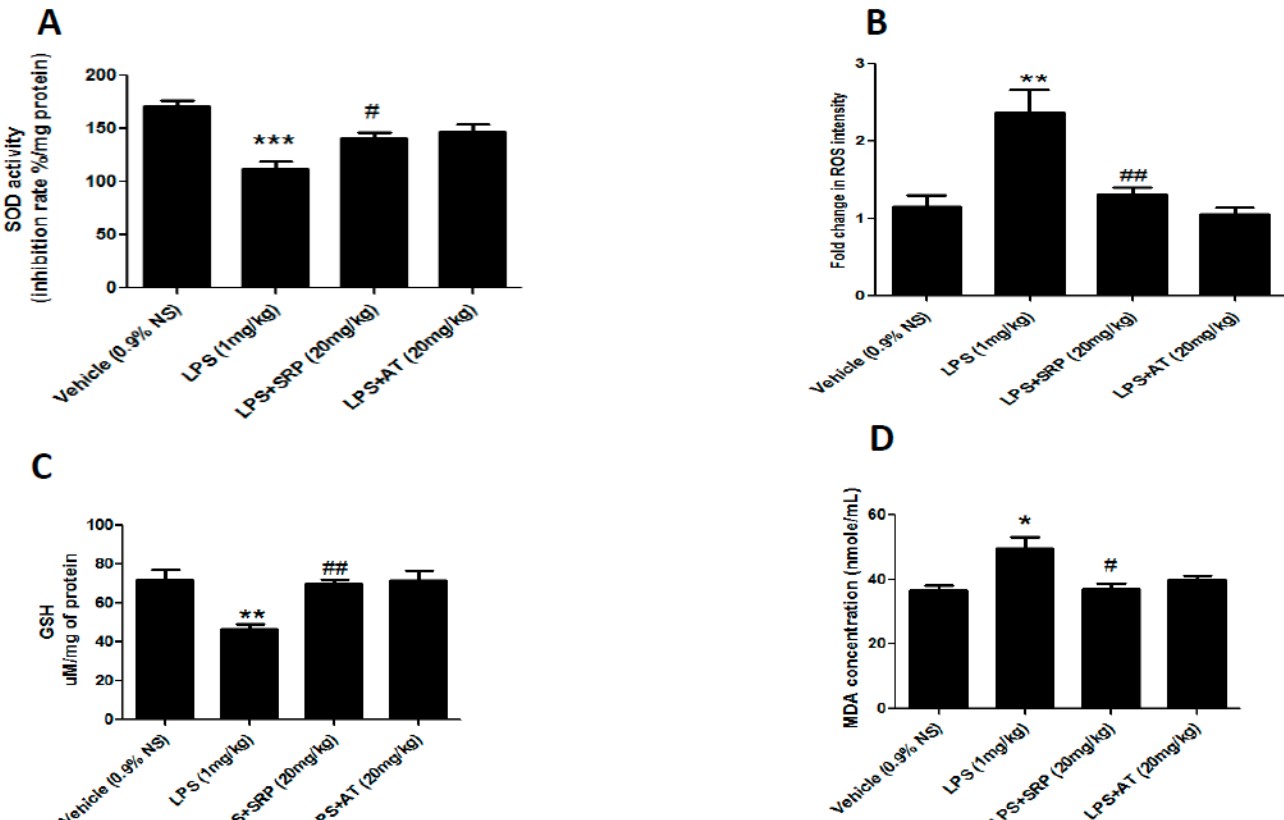

**Figure 1.** The effect of SRP on oxidative stress biomarkers. Column figure shows the oxidative stress biomarkers measured in aortic homogenates of vehicle control (0.9% NS), LPS-, LPS+SRP-, and LPS+AT-treated animals. (**A**) Superoxide Dismutase; (**B**) reactive oxygen species intensity; (**C**) glutathione activity; (**D**) malondialdehyde (MDA) represent mean ± SEM. n = 6 for each group. * represents significant values when compared to controls at $p < 0.05$, **#** represents significant values when compared to LPS-treated animals at $p < 0.05$, *** represents significant values when compared to control group at $p < 0.05$, ** represents significant values when compared to control group $p < 0.05$, **##** represents significant values when compared to LPS-treated group at $p < 0.05$.

### 3.3. Effect on Production of MCP-1 in Aorta

Immunohistochemical staining showed that compared with the control group, the LPS group had higher overexpressed protein levels of MCP-1, which were mostly ameliorated after serratiopeptidase treatment (Figure 3).

### 3.4. Effect on Vascular Endothelium in Aorta

H&E stain presented the integrity of the vascular endothelium, consisting of an unbroken endothelial monolayer with regularly shaped and arranged endothelial cells in the normal group. However, the endothelial layer exhibited remarkable histopathological changes in the LPS group, showing cellular edema and the partial exfoliation of endothelial cells. The histopathological change in the vascular endothelium was improved in the LPS + SRP group compared with the LPS group (Figure 4A–D).

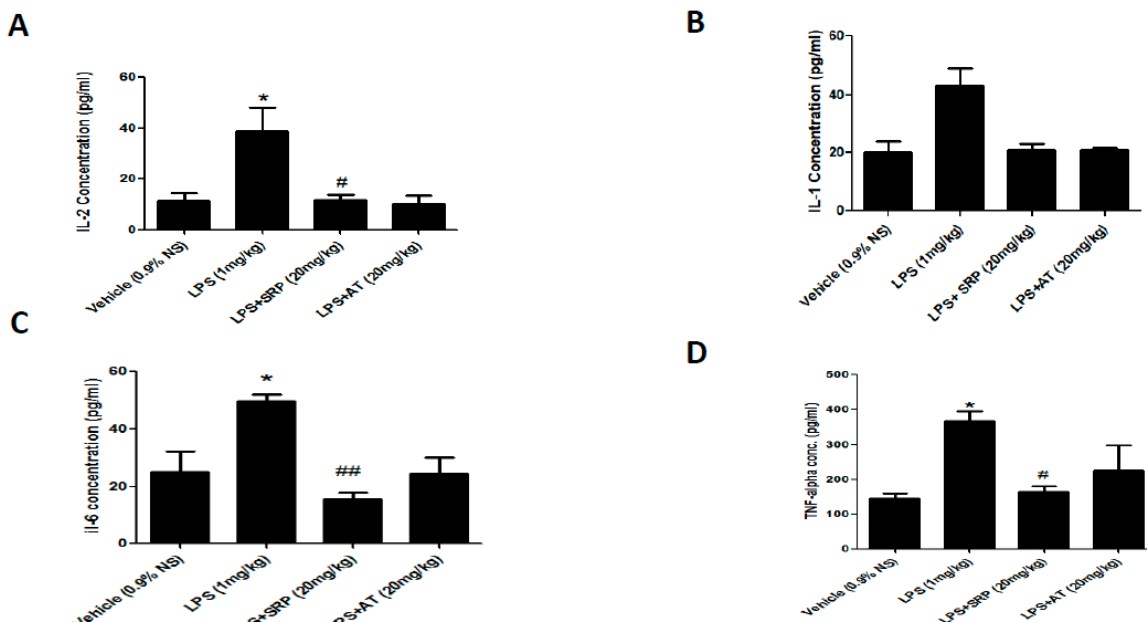

**Figure 2.** The effect of Serratiopeptidase on inflammatory biomarkers. Column figure shows the inflammatory biomarkers measured in whole aorta homogenates of vehicle control (0.9% NS), LPS-, LPS+SRP-, and LPS+AT-treated animals. (**A**) Interleukins-2; (**B**) interleukins-4; (**C**) interleukins-6; (**D**) TNF-alpha. Data represent mean ± SEM. n = 6 for each group. * represents significant values when compared to controls at $p < 0.05$, **#** represents significant values when compared to SRP-treated animals at $p < 0.05$, **##** represents significant values when compared to LPS-treated group at $p < 0.05$.

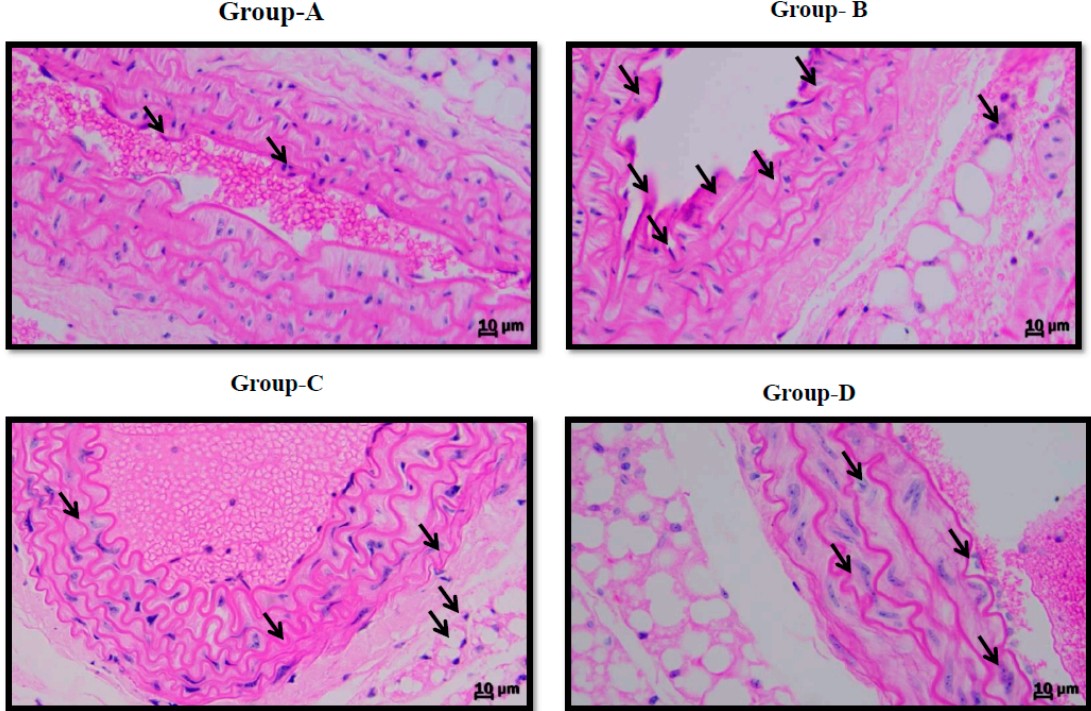

**Figure 3.** Aortic representative slices (5 µm-thick) stained with H&E. (**A**) Control group. (**B**) Endothelial cells in LPS-induced mice had significant damage, including disordered organization, indistinct borderlines, and partial deletion (arrow). (**C**) LPS + SRP-treated mice, who suffered less damage than LPS-treated mice. (**D**) LPS+AT-treated mice, who suffered less damage than LPS mice. Scale bars indicate 10 µm. (n = 6). Original magnifications—100×.

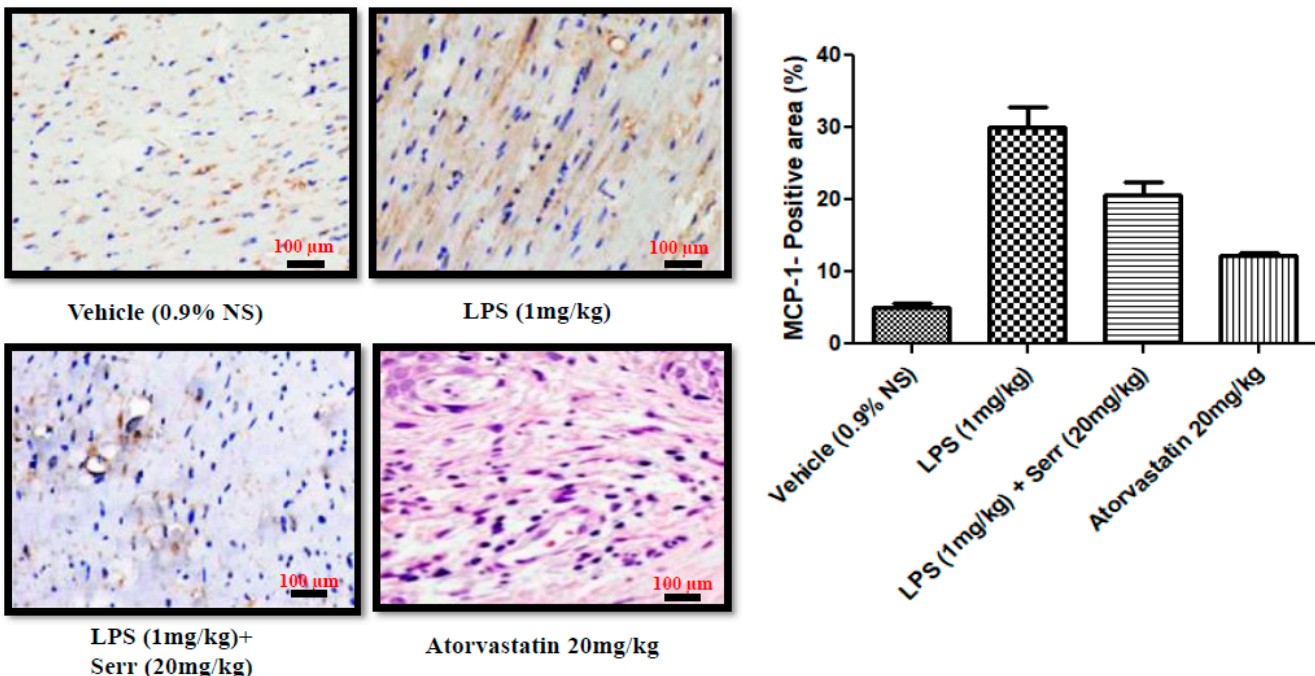

**Figure 4.** Immunohistochemistry analysis. MCP-1 expression of the heart ventricular sections from each mouse (20×, scale bar: 50 μm; n = 6 per group, at least 10 randomly selected heart sections per mouse). Data are presented as mean standard deviation. One-way ANOVA was applied.

## 4. Discussion

Cardiovascular disorders are widely regarded as the "first enemy" of human life [34]. Epithelial tissue is vital for providing the safety of the human physiology of the vascular system. The first cells to respond to excitotoxicity are endothelial cells as they serve as a bridge between the blood flow and the vessel wall. These cells are essential for the biological mechanisms of the vascular system [35]. Vascular inflammation (particularly inflammatory conditions) is linked to the pathophysiology mechanism of CVD formation and progression. Inflammatory conditions such as rheumatoid arthritis, Bechet's disease, and inflammatory bowel disease all enhance the chance of developing CVD in individuals. Anti-inflammatory drug use can enhance endothelial function in the resistance and conduit arteries, supporting the idea that inflammation plays a significant role in predicting cardiovascular disease (CVD) in inflammatory disorders [36]. As a result, developing innovative therapeutic approaches and strategies to preserve vascular endothelial function homeostasis in the prevention and treatment of CVDs is critical. Massive interest has been given to SRP's anti-inflammatory properties. Inflammatory response has been linked to the onset and progression of several disorders, and monocyte/macrophage infiltration-mediated organic inflammatory processes are important elements in the prolonged development of organic inflammation. The vasculature plays a crucial role in preserving circulatory equilibrium and halting the progression of cardiac disorders. A disturbance in its function, however, can cause ED, which, if addressed, might develop into the creation of atherosclerotic lesions and subsequent cardiac events. In order to detect capillary irregularities and maybe keep track of treatments and therapies that can enhance endothelial dysfunction and reduce the risk of heart disease, it is crucial to measure endothelial function in individuals at risk for the illness. A substantial number of monocytes and macrophage cells can invade inflamed tissues when exposed to MCP-1, a CC chemokine. Several inflammatory circumstances connected to monocyte recruitment result in the expression of MCP-1. By combining in vitro and in silico investigations of several inflammatory mediators, Ziraldo et al. found

that the chemokine MCP-1 serves as a primary coordinator of the inflammatory response of stressed endothelium. The activation of proinflammatory pathways by LPS is thought to contribute to the development of atherosclerotic plaque and vascular inflammation, which in turn contributes to cardiovascular disorders. In particular, persistent diseases brought on by Gram-negative bacteria raise the risk of atherosclerosis even in low-risk people who do not have typical vascular possible risks.

SRP, a serine protease with a molecular weight of 60 kDa, has been extensively studied for its anti-inflammatory properties. Many disorders, including arthritis, sinusitis, inflammatory bowel disease (IBD), and bronchitis, have been observed to benefit from enzyme therapy [37]. SRP has been shown to be useful in the treatment of ortho-dental inflammatory syndrome by Khateeb et al. [38]. Furthermore, because the enzyme metabolizes non-living tissues while leaving living tissue intact, it might be useful for eliminating fatty material, cholesterol, cellular waste products, calcification, and fibrin accumulation on the interior of the capillaries, and it could be utilized as a therapeutic candidate in various cardiovascular disorders. SRP's fibrinolytic (clot removal) action may also aid with thicker blood, in cases of elevated risk of stroke, and phlebitis/thrombophlebitis [37]. Furthermore, the protective effects and therapeutic potential of SRP on endothelium damage and the actual mechanism remain unknown.

In this work, we have assessed how SRP protected vascular endothelium in mice with LPS-induced acute inflammatory damage.

The endothelium is the fundamental layer that protects intravascular and extravascular cells/tissues as well as regulating vascular tone in both normal and diseased conditions [39]. To assess the preventive properties of SRP on vascular endothelial cells, we measured cytokine production levels in LPS-induced aortic tissue. As shown in Figure 2, compared with the control group, SRP promoted a decrease in oxidative stress and pro-inflammatory cytokines such as IL-2, IL-4, IL-6, and TNF-$\alpha$. Similarly, the production of oxidative stress in the thoracic aorta was significantly reduced in the treatment groups (Figure 1). SRP also significantly decreased the lesions formed on the thoracic aorta (Figure 4). Overall, the findings revealed that SRP protected the vascular endothelium against LPS-induced damage in animals by maintaining the equilibrium of cytokine production and that this action of SRP might be ascribed to MCP-1 regulation. MCP-1 is a cytokine belonging to the beta (C-C) family that attracts blood monocytes and T-cells and enhances their transepithelial motility [40]. MCP-1, along with other chemokines, engages in a cascade of cellular processes that seriously impact the endothelial cells as well as the adjacent tissues during septicemia [41].

Our histological data also demonstrated significant inflammatory lesion development post-LPS treatment, highlighting its role as an endothelium dysfunction agent and developing features such as proatherogenic cytokines, as shown in Figure 4. Furthermore, a potential protective effect of SRP was observed in our research work as compared to the controls. SRP lowered MCP-1 production in LPS-induced mouse aorta tissues, as seen in Figure 3, implying that SRP controlled MCP-1 production. Lipopolysaccharide has been demonstrated to cause endothelial cell mortality as well as changes in the endothelium barrier function [42,43]. We demonstrated that SRP seems to have the beneficial ability to reduce LPS-induced inflammation and ROS generation in the vascular endothelium using the LPS-induced endothelium insufficiency paradigm. SRP reduced the development of all pro-inflammatory cytokines and chemokines studied, including IL-1, TNF-$\alpha$, IL-6, and MCP-1. Certain mediators play key roles in inflammatory conditions. Epithelial cells are increasingly recognized as a significant source of multifunctional cytokine production, with endothelial-derived cytokines implicated in hematopoiesis, immunological response, coagulation, as well as other activities [44]. As a consequence, SRP administration greatly lowers LPS-induced cytoplasmic ROS and MDA levels, which are lipid peroxidation products [45], in the vascular endothelium. The current findings indicate that MCP-1 plays a role in vascular inflammation caused by LPS in the mouse aorta. These findings are based on the following observations: (1) LPS enhanced the production of the protein MCP-1,

which SRP inhibited; and (2) MCP-1 inhibition decreased LPS-induced oxidative stress and vascular inflammation.

## 5. Conclusions

In conclusion, our work demonstrates that SRP can attenuate the elevated inflammatory biomarkers (MCP-1, IL-2, IL-4, IL-6, and TNF-$\alpha$), the production of MDA, ROS intensity, CATx, and GSH impact in the setting of LPS-induced vascular inflammation. SRP inhibits inflammation and oxidative stress through the endothelium MCP-1. We concluded that SRP has the ability to modulate vascular inflammation. Further studies will be required to explore the detailed mechanisms of the cardio-protective effects produced by SRP. These results suggest that serratiopeptidase may be a therapeutic agent for vascular inflammation in cardiovascular diseases.

**Author Contributions:** S.S. (Sanjiv Singh) and V.R.: conceptualization, methodology, data analysis, draft preparation, reviewing final form for submission. V.Y.: conceptualization, methodology, data curation, original draft preparation. S.S. (Satyam Sharma): methodology and original draft preparation A.K.: draft preparation and methodology designing. All authors have read and agreed to the published version of the manuscript.

**Funding:** This research received funding from National Institute of Pharmaceutical Education and Research (NIPER) in Hajipur.

**Institutional Review Board Statement:** The Institutional Animal Ethics Committee (IAEC) of the National Institute of Pharmaceutical Education and Research (NIPER) in Hajipur, India, approved the research protocol, and all experiments followed the guidelines set forth by the Committee for the Purpose of Control and Supervision of Experiments on Animals (CPCSEA) in New Delhi, India. IAEC Certificate number NIPER-H/IAEC/09/21.

**Data Availability Statement:** Not applicable.

**Conflicts of Interest:** The authors declare no conflict of interest.

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
