# Peer review of "Serratiopeptidase Attenuates Lipopolysaccharide-Induced Vascular Inflammation by Inhibiting the Expression of Monocyte Chemoattractant Protein-1"

_cimb, doi:10.3390/cimb45030142_

Round 1

Reviewer 1 Report

Authors investigated the anti-inflammatory effect of SRP on LPS-induced vascular inflammation in mice. They concluded that SRP attenuates LPS-induced vascular inflammation by inhibiting the expression of MCP-1. But there are still some questions should be explained in the manuscript.

1. In Section 2.1, mice were treated with Atorvastatin alone or combined with LPS?

2. Why IL-2, IL-4, IL-6 and TNF-α were detected?

3. Why oxidative stress and MCP-1 were detected at the same time? Is there any relationship between the oxidative stress indexes and the four cytokines/MCP-1?

Author Response

Reviewer report – 1

S.No.

Comments

Answer

1.

In Section 2.1, mice were treated with Atorvastatin alone or combined with LPS?

In section 2.1, mice were treated with LPS+ Atorvastatin

(Corrected in manuscript)

2.

Why IL-2, IL-4, IL-6 and TNF-α were detected?

In this research study our intention was to prove LPS having ability to induced vascular inflammation in mouse and also evaluated the anti-inflammatory response of SRP in the mouse model. As per the previous strong research findings LPS and other pesticides have ability to enhance the secretions of IL-2, IL-4, IL-6 and TNF-α as well as overexpression of MCP-1 in cellular and molecular level and highly associated with vascular inflammation.

3.

Why oxidative stress and MCP-1 were detected at the same time? Is there any relationship between the oxidative stress indexes and the four cytokines/MCP-1

Various research findings and data strongly recommended that there is strong association between oxidative stress elevation can leads the cytokines productions in the cellular level. Our study demonstrates that the levels of IL-2, IL-4, IL-6, TNF-α and MCP-1 were significantly increased in the severe in the LPS treated mouse. Among the tested correlations between the oxidative stress parameters and the cytokines, the strongest evidence shown between oxidative stress index and interleukins. The presented results will contribute to support the evidences that the oxidative stress elevations can leads to cytokine production in LPS treated mice.

Reviewer 2 Report

The authors investigated about the effects of Serratiopeptidase in vascular inflammation caused by LPS in BALB/c mice. Overall the topic could be interesting but some details could be improved.

 I recommend that the paper be accepted with major revision:

a)   The authors should extend the background section in the abstract.

b)     The authors should mentioned in the abstract more details about model used.

c)        In the introduction section, little previous evidence is provided about the importance of vascular damage. Incorporating comparisons with other studies would increase the strength of the paper. Please refer to doi: 10.2174/0929867328666210329120213; 10.1371/journal.pone.0240669; 10.3390/ijms222111388.

d). The authors should add the number of mice used in their study and how they choose the number.

e)  The quality of all figures is very poor, in particular Figure 3. The figures should be improved.

f)     The authors should better emphasize the conclusion

 g)   There are some minor grammar issues that should be fixed in order to aid the accessibility of the results to the reader.

Author Response

Reviewer report – 2

S.No.

Comments

Answer

1.

The authors should extend the background section in the abstract.

 (Added specific points as per instruction of MDPI editorial board in abstract and also Corrected in manuscript)

2.

The authors should mentioned in the abstract more details about model used

(The detail about animals and related data has been added and corrected in abstract section)

3.

In the introduction section, little previous evidence is provided about the importance of vascular damage.

Incorporating comparisons with other studies would increase the

strength of the paper. Please refer to doi:10.2174/0929867328666210329120213;

10.1371/journal.pone.0240669; 10.3390/ijms222111388.

(The detail about animals and previous evidences of studies and recommended paper has been preferred and added in manuscript in the introduction section)

4.

The authors should add the number of mice used in their study and how they choose the number.

6 animals in each group total 24 animal has been used

5.

The quality of all figures is very poor, in particular Figure 3. The figures should be improved.

We have replaced and did the modifications as per the suggestions

6.

The authors should better emphasize the conclusion

We have emphasized the data in conclusion part and followed the proper instruction

7.

There are some minor grammar issues that should be fixed in order to aid the accessibility of the results to the reader.

All the grammar’s and languages has been fixed as per the manuscript demand and editor instructions

Reviewer 3 Report

The authors did a great work and the manuscript is well written despite some minor issues.

1. Line 2-3: Please consider to not use abbreviations in the title of the manuscript.

2. Line 55: “Microorganisms endotoxin”. This is not understandable. Please consider to rewrite. Maybe just “entotoxins”.

3. Lines 69-70: …., in an inflammation model. I would suggest to remove the statement.

4. Line 71: … of recognize… Please change to “…of recognizing….”

5. Line 139: “3. Result:” please remove “:” “

6. References should be updated by including current references.

Author Response

Reviewer report – 3

S.No.

Comments

Answer

1.

Line 2-3: Please consider to not use abbreviations in the title of the manuscript.

 We have removed the abbreviations and added the full name of all specific words in title.

2.

Line 55: “Microorganisms endotoxin”. This is not understandable. Please consider to rewrite. Maybe just “entotoxins”.

We have removed and replaced the words as per the instruction.

3.

Lines 69-70: …., in an inflammation model. I would suggest to remove the statement.

Removed the statement and corrected manuscript

4.

Line 71: … of recognize… Please change to “…of recognizing….”

Line 71: … of recognize… replaced to “…of recognizing….” As per the instructions

5.

Line 139: “3. Result:” please remove “:” “

Modified and corrected

6.

References should be updated by including current references.

We have  added the several new references related to study

Round 2

Reviewer 1 Report

It is acceptable now.